# Single-layer graphene membranes by crack-free transfer for gas mixture separation

Shiqi Huang[1], Mostapha Dakhchoune[1], Wen Luo[2], Emad Oveisi [3], Guangwei He[1], Mojtaba Rezaei[1], Jing Zhao[1], Duncan T.L. Alexander[3], Andreas Züttel[2], Michael S. Strano[4] & Kumar Varoon Agrawal [1]

The single-layer graphene film, when incorporated with molecular-sized pores, is predicted to be the ultimate membrane. However, the major bottlenecks have been the crack-free transfer of large-area graphene on a porous support, and the incorporation of molecular-sized nanopores. Herein, we report a nanoporous-carbon-assisted transfer technique, yielding a relatively large area (1 mm$^2$), crack-free, suspended graphene film. Gas-sieving ($H_2/CH_4$ selectivity up to 25) is observed from the intrinsic defects generated during the chemical-vapor deposition of graphene. Despite the ultralow porosity of 0.025%, an attractive $H_2$ permeance (up to $4.1 \times 10^{-7}$ mol m$^{-2}$ s$^{-1}$ Pa$^{-1}$) is observed. Finally, we report ozone functionalization-based etching and pore-modification chemistry to etch hydrogen-selective pores, and to shrink the pore-size, improving $H_2$ permeance (up to 300%) and $H_2/CH_4$ selectivity (up to 150%). Overall, the scalable transfer, etching, and functionalization methods developed herein are expected to bring nanoporous graphene membranes a step closer to reality.

[1] Laboratory of Advanced Separations (LAS), École Polytechnique Fédérale de Lausanne (EPFL), Sion 1950, Switzerland. [2] Laboratory of Materials for Renewable Energy (LMER), École Polytechnique Fédérale de Lausanne (EPFL), Sion 1950, Switzerland. [3] Interdisciplinary Centre for Electron Microscopy (CIME), École Polytechnique Fédérale de Lausanne (EPFL), Lausanne 1015, Switzerland. [4] Department of Chemical Engineering, Massachusetts Institute of Technology, Cambridge, 02139 MA, USA. Correspondence and requests for materials should be addressed to K.V.A. (email: kumar.agrawal@epfl.ch)

Atom-thick graphene film is the thinnest possible molecular barrier and therefore incorporated with molecular-sized pores, it can be regarded as the ultimate membrane[1]. Several molecular simulations have shown that the two-dimensional nanopores in graphene can yield high gas permeance, orders of magnitude higher than that attainable with the conventional membranes[2–11]. Such high-flux membranes can considerably reduce the needed membrane area for separating a volume of a molecular mixture, addressing the problem of scale-up, a longstanding issue with the inorganic membranes. The thermal and chemical robustness and the high mechanical strength of the graphene lattice, even with a porosity as high as 5%[12,13], makes it highly attractive for the gas separation. Recently, several etching methods for incorporating sub-nanometer pores in graphene lattice have been developed, leading to promising sieving performances for liquids and dissolved ions[14–17]. However, the demonstration of gas mixture separation from single-layer graphene membrane has remained a challenging task[18,19]. A proof-of-concept was demonstrated by measuring the deflation rate of a bilayer graphene microballoon, where pores were incorporated by multiple ultraviolet treatment[20]. In general, the molecular transport studies through single-layer graphene have been primarily carried out on micron-sized domains, attributing to the poor scalability of the micromechanical exfoliation, and the challenging transfer of the chemical-vapor deposition (CVD) derived graphene. Celebi et al. reported a $2500 \, \mu m \mu m^2$-sized bilayer graphene membrane by masking cracks in a graphene film by another graphene film[21]. Using focused-ion beam, an array of nanopores (>7.6 nm in diameter) were incorporated leading to an effusive gas transport. Recently, a centimeter-scale single-layer graphene membrane hosting molecular-sized pores was reported, however, the cracks generated during the transfer limited the separation selectivity close to that expected from the Knudsen diffusion ($H_2/CH_4$ and $He/SF_6$ selectivities of 3.2 and 8.0, respectively, were reported)[18]. Overall, the demonstration of gas mixture separation from sufficiently-scaled single-layer graphene membrane has remained elusive. To develop graphene membranes, one needs to (a) transfer large-area graphene onto porous supports without generating cracks and tears, and (b) generate molecular-sized pores with a narrow pore-size-distribution (PSD). Development of such method would also allow one to study the gas transport mechanism (activated vs. surface vs. Knudsen transport), and effect of the competitive adsorption through the graphene nanopores.

Herein, we report a novel nanoporous-carbon-assisted graphene transfer technique that enables transfer of relatively large area (1 mm²) single-layer CVD graphene onto a macroporous support (pore-opening of 5 μm) without generating cracks or tears, allowing observation of gas-sieving from the intrinsic defects of CVD graphene. An attractive $H_2$ permeance (up to $4.1 \times 10^{-7} \, mol \, m^{-2} \, s^{-1} \, Pa^{-1}$) is obtained despite the ultralow-density of the intrinsic defects (porosity of 0.025%). An activated gas transport is observed with an average activation energy for $H_2$ transport across eight membranes being 20.2 ± 2.7 kJ/mole. The molecular-sized intrinsic defects yield an attractive separated selectivity, including those from the mixed gas feed ($H_2/CH_4$ separation factor up to 18). The membrane performance remains stable during several heating and cooling cycles (25–150 °C), and at least up to 7 bar of the transmembrane pressure difference. Finally, in the pursuit to increase the density of gas-selective pores, we also report an ozone functionalization-based etching and pore-modification chemistry, increasing the nanopore density and/or reducing the effective pore-size. A combination of higher selectivity, higher permeance, and higher selectivity/higher permeance is observed.

## Results

**Crack-free transfer of CVD graphene.** CVD derived single-layer graphene is well-suited for the fabrication of large-area membranes attributing to the scalability of the CVD process[22,23]. Post-CVD, graphene needs to be transferred from the catalytic metal foil to a porous support for the membrane fabrication. However, the conventional transfer techniques invariably introduce cracks and tears in the graphene film[24], and therefore, so far the suspended, crack- and tear-free, single-layer graphene membranes have been limited to a few μm² in area[14,17,25]. Among several transfer techniques developed so far, the wet-transfer technique has been investigated the most attributing to its versatility allowing transfer of graphene on a wide-range of substrates[26–29]. Briefly, the exposed surface of graphene lying on a metal foil is coated with a sacrificial mechanically reinforcing polymer film (typically 100–200 nm thick poly(methyl methacrylate) (PMMA) film). Subsequently, the metal foil is removed by etching the metal in an etchant bath leaving the polymer coated graphene floating on the bath. Finally, the floating film is scooped on top of the desired substrate, and the polymer film is dissolved away to expose the surface of graphene. This wet-transfer process has been proven to be quite successful in fabricating graphene-based devices on smooth non-porous substrates[24]. However, significant cracks and tears develop in the graphene film when a porous support is used, primarily because of a strong capillary force on the suspended graphene film during the solvent drying stage[30]. This issue can be mitigated if the mechanically reinforcing film is not removed, and yet somehow the graphene surface is exposed. Motivated by this, we developed a nanoporous carbon (NPC) film-assisted transfer method (Fig. 1), where at the end of the graphene transfer, the NPC film is left on top of the graphene film. Briefly, a solution of turanose and polystyrene-co-poly(4-vinyl pyridine) (PS-P4VP) was spin-coated on top of the as-synthesized CVD graphene. The block-copolymer film undergoes phase separation into hydrophobic and hydrophilic domains upon drying[31]. Subsequently, the film was pyrolyzed at 500 °C in the flow of $H_2/Ar$, leading to the formation of the NPC film on top of graphene. Scanning electron microscope (SEM) images of the NPC/graphene film on Cu, and transmission electron microscopy (TEM) images of the transferred NPC/graphene film on a TEM grid revealed that the NPC film was 100 nm thick, and comprised of 20–30 nm sized nanopores (Fig. 2a–c), which should expose at least 50% of the graphene surface. Selected area electron diffraction (SAED) of the composite NPC/graphene film (Fig. 2d), displayed the typical diffraction peaks of a suspended single-layer graphene, representing periodicities of 0.213 and 0.123 nm[32]. NPC film (Supplementary Note 1, Supplementary Table 1) contributed to the SAED with broad rings, a characteristic of the amorphous structure (Supplementary Figure 1). We could not find any area representing only the NPC film, indicating that graphene and NPC film bonded strongly during the pyrolysis step and that the graphene did not peel off from the NPC film during the etching of the metal foil. This is highly important for crack-free transfer of graphene, otherwise poor interactions of graphene with a support film can lead to severe cracks and tears during the transfer step[33]. The NPC coated graphene was transferred from the Cu foil to a custom-made macroporous support (porous area of 1 mm², Fig. 2e, f). The support was fabricated by laser drilling an array of 5 μm pores in a 50-μm-thick W foil[25] (Supplementary Note 2). Inspection of the transferred film by the SEM confirmed that there were no visible tears or cracks in the transferred film (Fig. 2g). Interestingly, even a macroscopic fold as shown in Fig. 2g did not break the membrane, making this process highly promising for the scale-up of single-layer graphene membrane.

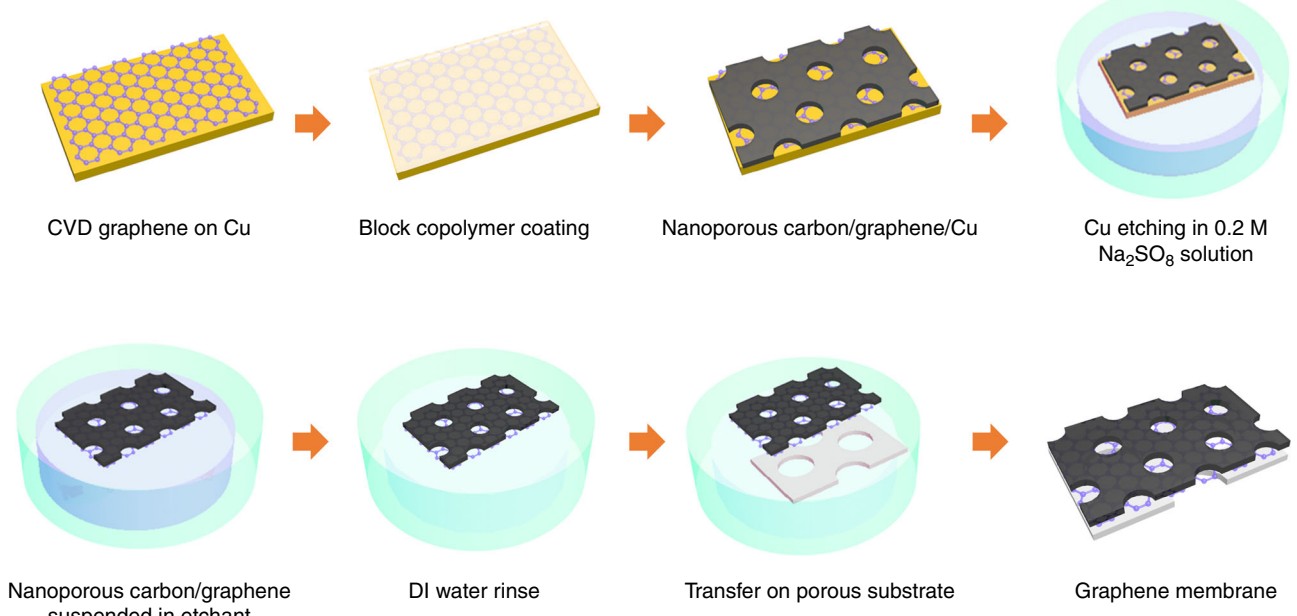

**Fig. 1** Schematic of fabrication of large-area graphene membrane by the nanoporous carbon (NPC) film-assisted transfer method. A block-copolymer solution was spin-coated onto the CVD graphene supported on the Cu foil; pyrolysis was conducted to form NPC film on top of graphene. The copper was etched by 0.2 M sodium persulfate, after which the floating graphene/NPC film was rinsed with DI water. Finally, the NPC/graphene film was transferred onto the porous tungsten support

**Gas transport through intrinsic defects of graphene**. Using scanning tunneling microscope (STM) directly on the as-synthesized graphene supported on the Cu foil, we recently imaged the low-density of intrinsic defects in the CVD graphene[25]. These defects are essentially molecular-sized pores (missing 10–16 carbon atoms), formed by etching of the graphene lattice in the presence of residual oxygen in the CVD chamber, and are promising for the gas separation. In the current study, the density of defects, estimated by the carbon amorphization trajectory[34] ($I_D/I_G$ of 0.07 ± 0.02, Fig. 2h, Supplementary Figure 4), was $5.4 \times 10^{10}$ cm$^{-2}$ corresponding to a porosity of 0.025%. A survey of the graphene lattice by aberration-corrected high-resolution TEM (HRTEM) revealed several sub-1-nm nanopores with a pore-density of $2.8 \times 10^{11}$ cm$^{-2}$ (Fig. 2i–k). This small disagreement between the HRTEM survey and the estimate from the amorphization trajectory is expected especially at a low defect density. Nevertheless, the successful crack-free transfer of the CVD graphene allowed us to study the transport behavior of the intrinsic defects.

Graphene membranes were sealed in a homemade permeation cell using a metal face seal directly on top of the W support, ensuring a leak-proof measurement of the gas transport (details in the Methods). Typically, the feed side (a pure gas feed or a mixture feed) was pressurized to 1.5–7.0 bar, whereas the permeate side connected to a pre-calibrated mass spectrometer (MS) was maintained at 1 bar with an argon sweep (Supplementary Figure 2). Temperature of the membrane was varied between 25–250 °C. Single-component gas transport study from eight separate membranes revealed H$_2$ permeance in the range of $5.2 \times 10^{-9}$–$7.2 \times 10^{-8}$ mol m$^{-2}$ s$^{-1}$ Pa$^{-1}$ (15–215 gas permeation units, GPU) with H$_2$/CH$_4$, H$_2$/CO$_2$, and He/H$_2$ ideal selectivities ranging between 4.8–13.0, 3.1–7.2, and 0.7–2.0, respectively, at 25 °C (Fig. 3a–d, Supplementary Table 9–11). The H$_2$ permeance corresponded to a permeation coefficient of $1.0 \times 10^{-23}$–$1.3 \times 10^{-22}$ mol s$^{-1}$ Pa$^{-1}$ based on the defect density of $5.4 \times 10^{10}$ cm$^{-2}$. This permeation coefficient is consistent with that of Bi-3.4 Å membrane reported by Koenig et al. ($4.5 \times 10^{-23}$

mol s$^{-1}$ Pa$^{-1}$)[20]. The H$_2$/CH$_4$ selectivity was lower than that from Bi-3.4 membrane[20], indicating a wider PSD of intrinsic defects in CVD graphene, compared to PSD from pores incorporated in micromechanically exfoliated graphene. Based on the achieved H$_2$/CH$_4$ selectivities, the estimated percentage of larger nanopores yielding non-selective effusive gas transport is less than 25 ppm (refer to the Supplementary Note 4 and Supplementary Table 8 for more details). Interestingly, the H$_2$/CO$_2$ selectivity was higher than that of the Bi-3.4 membrane where a selectivity of ca. 1.5 was reported. Membrane M8 displayed the best molecular sieving performance and was the only membrane displaying He/H$_2$ selectivity greater than 1, implying that the mean pore-size in M8 was less than the kinetic diameter of H$_2$ (0.289 nm).

The graphene membranes did not rupture during heating to up to 250 °C. The permeance of He, H$_2$, CO$_2$ and CH$_4$ increased with temperature, indicating that its transport was in the activated transport regime. At 150 °C, the H$_2$ permeance increased to $3.3 \times 10^{-8}$ – $4.1 \times 10^{-7}$ mol m$^{-2}$ s$^{-1}$ Pa$^{-1}$ (100 - 1220 GPU), with H$_2$/CH$_4$, and H$_2$/CO$_2$ selectivities increasing to 7.1–23.5 and 3.6–12.2, respectively. We note that this H$_2$/CH$_4$ separation performance from single-layer graphene with 0.025% porosity approaches the 2008 Robeson upper bound for polymers[35] (assuming 1-μm-thick selective skin layer of the polymer membrane, Supplementary Figure 6). To understand the transport behavior, the activation energy for gas diffusion across the nanopores was extracted from the temperature-dependent gas flux using an adsorbed phase transport model developed using the concepts of adsorption and diffusion[36–39] (Eq (1), refer to Supplementary Note 3 for details).

$$\text{Flux} = C_o A_{act} A_{sur} \exp\left(-\frac{(E_{act} + \Delta E_{sur})}{RT}\right)(f(P_A) - f(P_R)) \quad (1)$$

where

$$f(P_x) = \frac{P_x}{1 + A_{sur}\exp\left(\frac{-\Delta E_{sur}}{RT}\right)P_x}$$

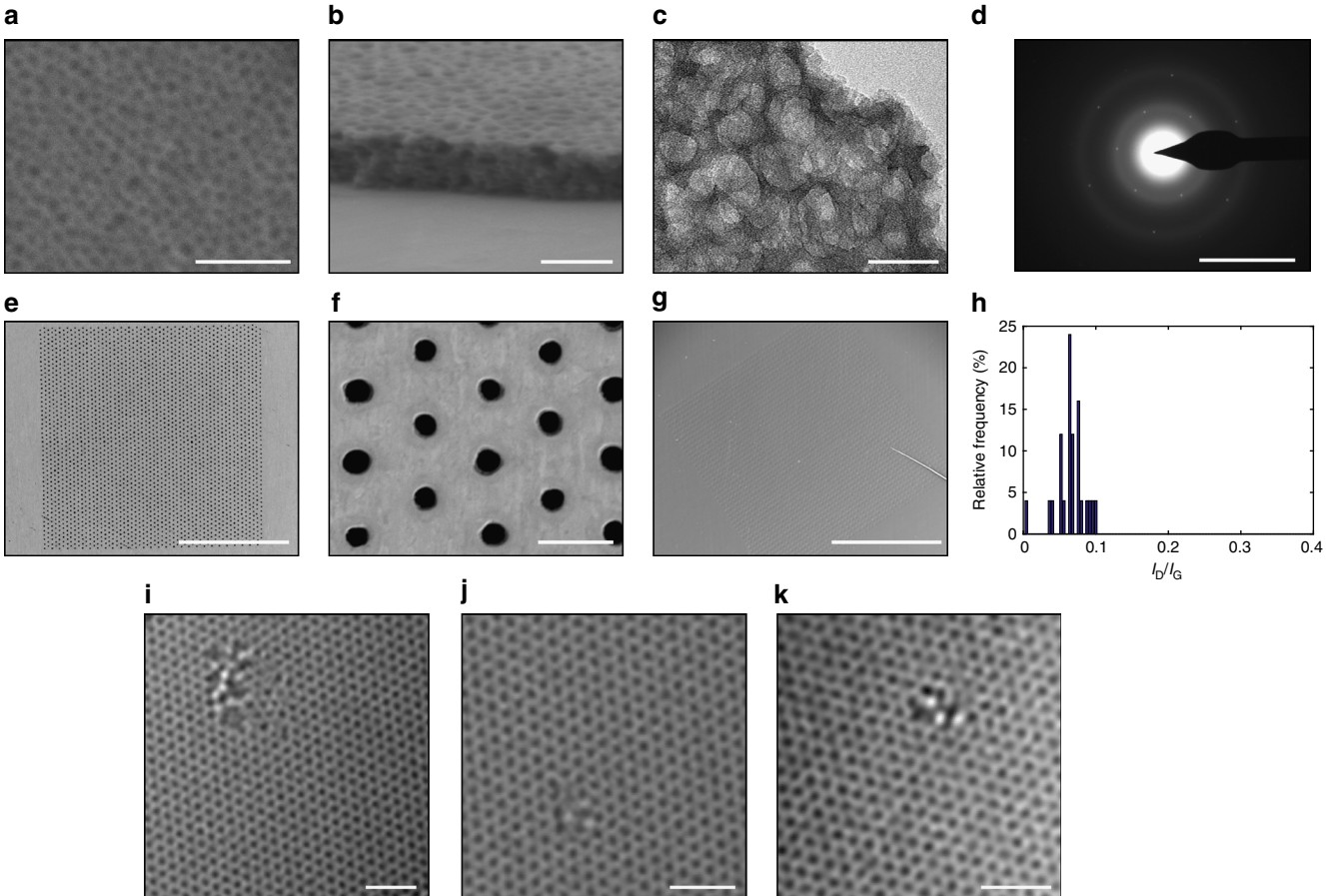

**Fig. 2** Synthesis, transfer, and characterization of low-pressure chemical-vapor deposition (LPCVD) derived graphene. **a** Scanning electron microscopy (SEM) image of the nanoporous carbon (NPC) film coated on top of graphene. **b** Cross-sectional SEM image of the composite NPC film and graphene. **c** Transmission electron microscopy (TEM) image of the composite NPC film/graphene. **d** The electron diffraction pattern from the composite film shown in **c**. **e** SEM image of porous tungsten support. **f** SEM image of porous tungsten support. **g** SEM image of the transferred graphene on the tungsten support. **h** Histogram of $I_D/I_G$ from LPCVD graphene. **i–k** High-resolution TEM (HRTEM) images of the intrinsic defects in graphene lattice. The unprocessed raw images are shown in Supplementary Figure 3a-c. Scale bars in **a**, **b**, and **c** are 200, 100, and 50 nm, respectively. Scale bar in **d** is 10 nm$^{-1}$. Scale bars in **e**, **f**, and **g** are 500, 20, and 500 μm, respectively. Scale bars in **i**, **j**, and **k** are 1 nm

Here, $C_O$ is the pore-density, $E_{act}$ is the activation energy for the gas diffusion across the nanopores, and $\Delta E_{sur}$ is the adsorption energy of gas on to the graphene nanopore. $A_{act}$ is the pre-exponential factor for the gas diffusion across the nanopores. $A_{sur}$ is the pre-exponential factor for the adsorption event, representing changes in the overall entropy. $T$ is the temperature, and $P_A$ and $P_R$ are the gas partial pressures on the feed and permeate sides, respectively. A comparison of $E_{act}$ for the four gases can indicate the ease with which the molecules diffuse across the nanopores, while a comparison of the pre-exponential factor, $C_oA_{act}A_{sur}$, can indicate the relative number of pores participating in the molecular diffusion. Average $E_{act}$ across eight membranes for He, $H_2$, $CO_2$, and $CH_4$ were 14.7 ± 3.2, 20.2 ± 2.7, 31.3 ± 2.8, and 25.8 ± 4.8 kJ/mol, respectively, increasing as a function of kinetic diameter (Fig. 3e, Supplementary Table 2). A slightly smaller $E_{act}$ for $CH_4$ in comparison to $CO_2$ can be explained by the fact that diffusion of $CH_4$ takes place from a smaller number of pores (average $C_oA_{act}A_{sur}$ for He, $H_2$, $CO_2$, and $CH_4$ were 1.5 × 10$^{-5}$, 2.6 × 10$^{-5}$, 3.8 × 10$^{-6}$, and 1.3 × 10$^{-6}$, respectively, Supplementary Table 3), assuming $A_{act}A_{sur}$ do not change significantly for $CO_2$ and $CH_4$[37]. The activation energy for $H_2$ was similar to that from hydrogen-functionalized pore-10 reported by Jiang et al. (0.22 eV)[3], providing an indication that the average pore in this study is close to that made from missing 10 carbon atoms,

which is supported by the HRTEM images (Fig. 2i–k) and our previous STM findings[25]. We note that while the gas permeance varied across the eight membranes, the activation energies for diffusion across the nanopores were consistent across the membranes. This indicates that, while the PSD was uniform across the membranes, the density of the intrinsic defects varied across the membranes.

Separation of a gas mixture can elucidate the contribution of competitive adsorption on the overall separation performance from the nanoporous graphene membrane. To the best of our knowledge, gas mixture separation through a single-layer graphene membrane has not been reported. Here, the large-area of the graphene membrane enabled measurements of He, $H_2$, $CO_2$, and $CH_4$ permeances from an equimolar gas mixture. The competitive adsorption[3,9,36,40] is expected to yield a reduced He/$H_2$, $H_2/CO_2$, and $H_2/CH_4$ separation factor (SF) compared to the corresponding ideal selectivities (IS). However, the separation factors were similar (He/$H_2$ and $H_2/CO_2$) or higher ($H_2/CH_4$) compared to the corresponding ideal selectivities (Fig. 4a–d). For example, for membrane M2, the $H_2/CH_4$ SF was higher than IS (10.8 vs. 5.7 at 25 °C and 12.2 vs. 11.2 at 150 °C), while the $H_2$ permeance (3.3 × 10$^{-8}$–2.2 × 10$^{-7}$ mol m$^{-2}$ s$^{-1}$ Pa$^{-1}$ between 25 and 150 °C) in the mixture case was similar to the single-component case (Fig. 4a). Similarly, the $H_2$ permeance did not

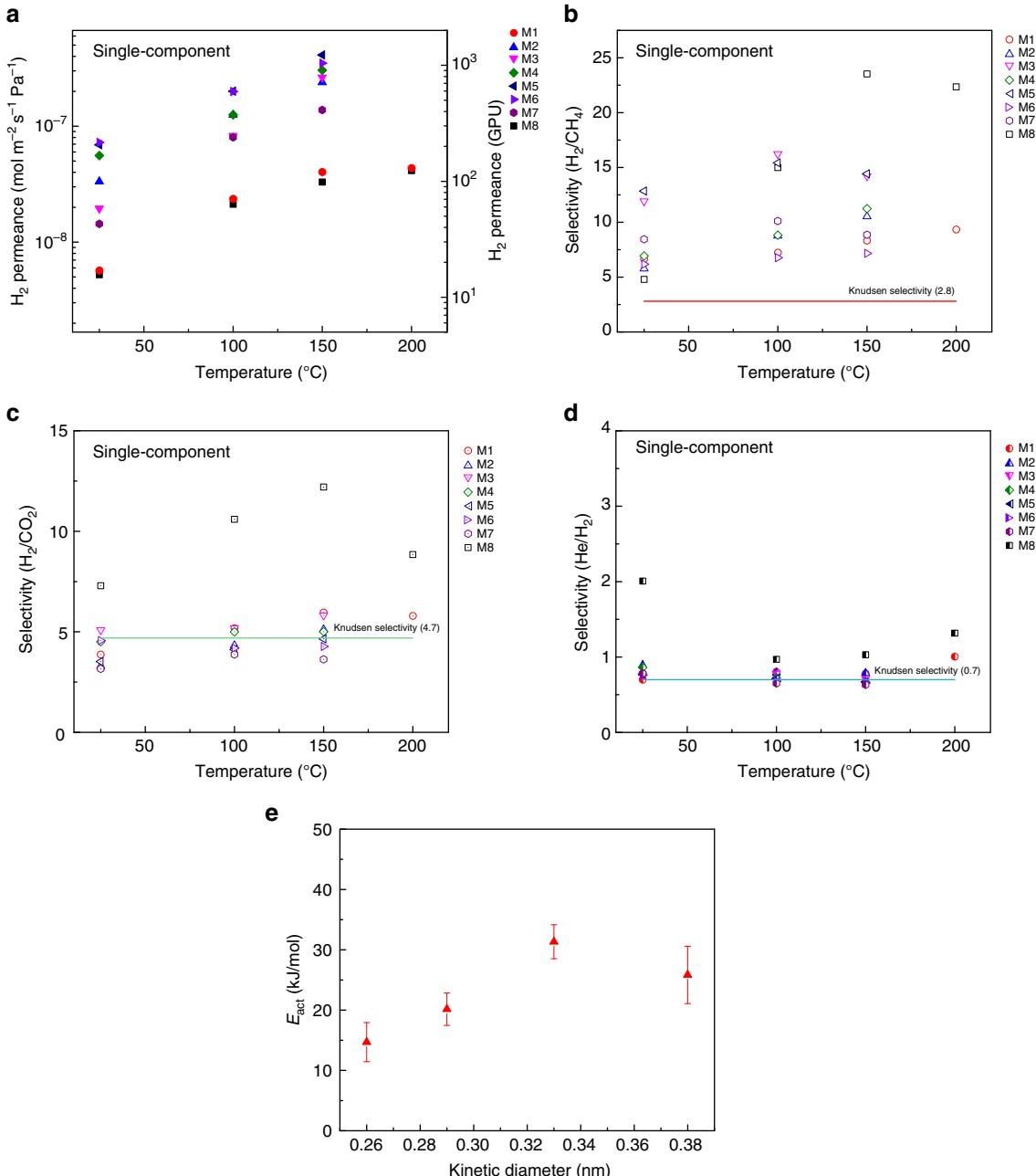

**Fig. 3** Gas separation performance of the intrinsic defects in graphene. **a** $H_2$ permeance across eight graphene membranes (M1–M8) as a function of temperature when using a single-component feed. **b–d** Ideal selectivities for various gas pairs from the eight membranes as a function of temperature; **b** $H_2/CH_4$, **c** $H_2/CO_2$, and (**d**) $He/H_2$. **e** Extracted activation energies (average across all eight membranes) are plotted as a function of the kinetic diameters of various gases

reduce for the membrane M3 for the mixture case, while the $H_2/CH_4$ SF increased to 18.0 from an IS of 14.2. For other membranes (M1, M4, M5, and M6), the $H_2$ permeance and the $H_2/CH_4$ selectivity in the mixture case were similar to those in the single-component case. These results indicate that the competitive adsorption of gases on the basal plane of graphene does not play a significant role in overall transport especially when the transport is in the activated regime, and when the feed pressure is moderate (up to 8 bar in this study). We expect the competitive adsorption to play a role at a higher feed pressure (30–50 bar, refer to Supplementary Note 3 and Supplementary Equation 36), which will be investigated in future studies.

The graphene membranes were thermally stable (Fig. 5a). In general, all membranes were stable at least up to 150 °C. For

instance, the performance of membrane M2, tested under three consecutive temperature cycles from 25 °C to 150 °C, did not change significantly. From cycle one to cycle three at 150 °C, the $H_2$ permeance decreased marginally ($3.3 \times 10^{-7}$ to $2.3 \times 10^{-7}$ mol m$^{-2}$ s$^{-1}$ Pa$^{-1}$), while the $H_2/CH_4$ selectivity increased marginally (8.3 to 10.5). Moreover, the graphene membranes were also stable at least up to 8 bar of mixture feed at 100 °C (permeate pressure 1 bar, Fig. 5b, c), where the $H_2$ permeance and the $H_2/CH_4$ separation factor did not change significantly.

**Ozone functionalization-based etching and pore-modification chemistry.** The porosity of graphene lattice yielding the attractive $H_2$ permeance was only 0.025%. Theoretically, the $H_2$ permeance

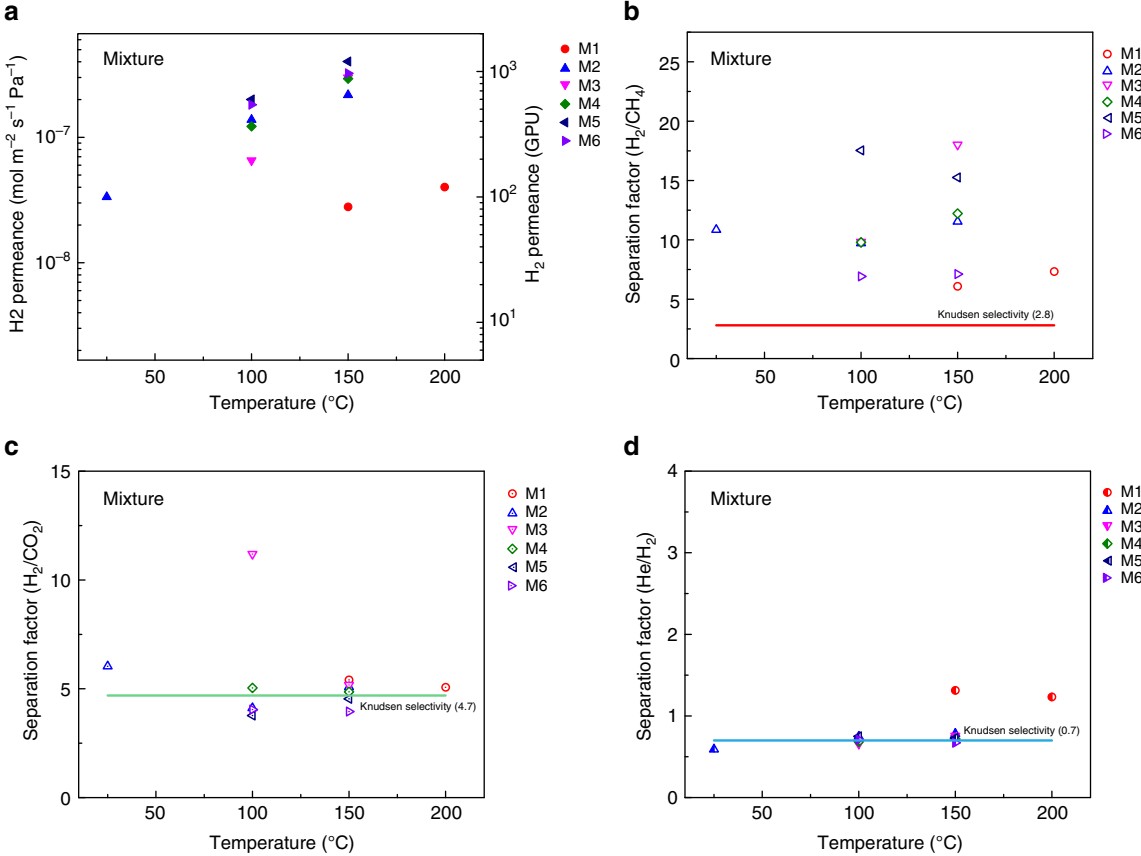

**Fig. 4** The gas mixture separation performance of the intrinsic defects in graphene. **a** $H_2$ permeance from six membranes (M1–M6) as a function of temperature when using an equimolar mixture gas feed. $H_2/CH_4$, $H_2/CO_2$, and $He/H_2$ separation factors as a function of temperature are shown in **b**, **c**, and **d**, respectively

can be further increased beyond $10^5$ GPU by increasing the defect density to $10^{12}$–$10^{13}$ cm$^{-2}$. On the other hand, the gas selectivity can be improved by constricting the nanopores. One way to achieve this is chemical functionalization of the pore-edge. Although, there are several potential chemical and physical routes to open pores in graphene, development of an in-situ etching method (inside membrane module) allowing a high degree of control is highly attractive. In this pursuit, we report a scalable ozone functionalization-based pore-etching and pore-edge-functionalization chemistry, improving the performance of the single-layer graphene membranes. We demonstrate that a controlled temperature-dependent oxidative functionalization of the graphene lattice with ozone-derived epoxy and carbonyl groups can be used to either etch molecular-sized pores in the CVD derived graphene or constrict the existing pores.

Oxidative treatment of graphene has been shown to incorporate sp$^3$-hybridized sites (epoxy and carbonyl groups) on the basal plane of graphene[41–43]. When the functionalization density is high, such as that in graphene oxide, one can introduce nanopores in the lattice by thermal annealing[44–47]. Typically, the functional groups migrate and rearrange forming larger groups (such as lactone), and finally desorb as CO or $CO_2$ leading to a vacancy[44]. Ozone, in the gas phase, can be conveniently used to oxidize graphene lattice. To understand the evolution of functionalization, CVD graphene supported on a Cu foil was exposed to ozone at various temperatures (25 °C, 80 °C, and 100 °C) and time (1 min to 7 min). The evolution of oxidative groups on graphene was probed by micro-Raman and X-ray photoelectron spectroscopy (XPS). The relative intensity of the D peak with respect to the G peak ($I_D/I_G$), which marks the extent of disorder

in graphene[48], increased from 0.07 to 4.0, while the intensity of the 2D peak decreased in intensity with the increasing reaction time and temperature, indicating that the sp$^3$-hybridized sites in graphene increased after ozone treatment (Fig. 6a, b). XPS indicated that C–O and C=O were the major functional groups on the functionalized graphene (Fig. 6c, d and Supplementary Figure 5). The number density of functional groups increased with the reaction temperature and time, in agreement with the Raman spectroscopy. In general, the density of C=O groups was higher than that of the C–O groups, even when the functionalization was carried out at room temperature for a short exposure of 2 min. At 100 °C, the degree of oxidation approached that of graphene oxide (35, 56, and 65% of the oxidized carbon lattice with exposure times of 2, 5, and 7 min, respectively). Overall, the functionalization was reproducible and was simple to implement. HRTEM images of the ozone-functionalized graphene (80 °C for 2 min) indeed revealed a higher pore-density ($4.2 \times 10^{11}$ cm$^{-2}$) compared to that in as-synthesized graphene (Fig. 6e–g). Moreover, the population of the sub-1-nm pores (87%) increased compared to that in the as-synthesized graphene (76%) (details in Supplementary Note 5).

To understand the effect of functionalization on the performance of graphene membrane, the graphene membranes were exposed to ozone, in-situ, in the permeation setup after probing the gas transport from the intrinsic defects. With this strategy, the gas transport before and after the functionalization could be compared. Overall, the separation performance of the graphene membranes improved, reflected by an increase in the $H_2$ permeance or an increase in the $H_2/CH_4$ selectivity or an increase in the permeance as well as the selectivity (Figs. 7 and 8). When

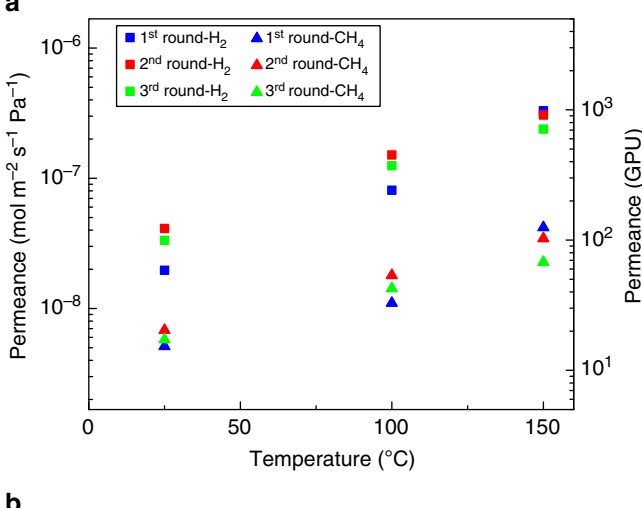

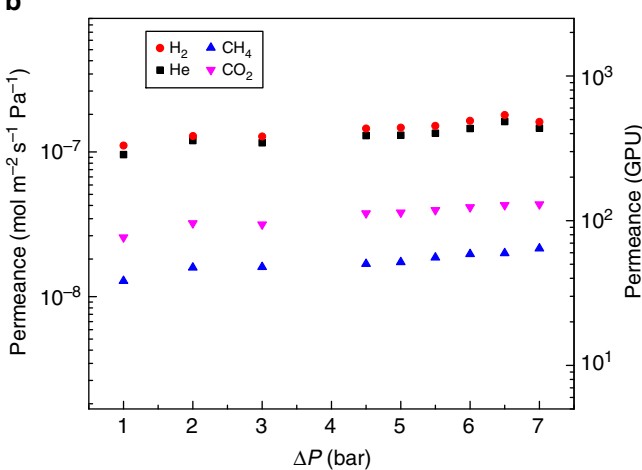

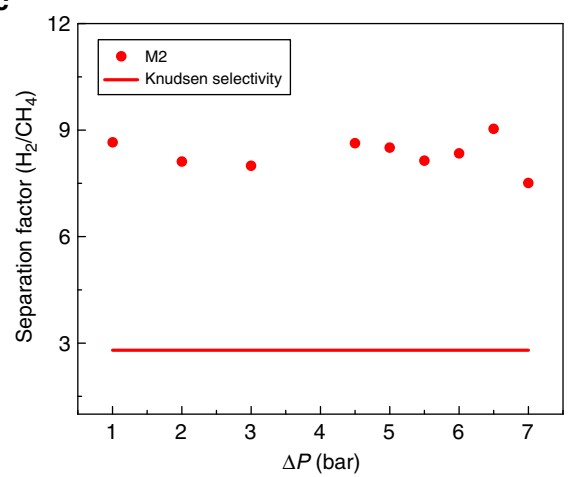

**Fig. 5** Stability test of the graphene membrane. **a** $H_2$ and $CH_4$ permeances of the membrane M2 with three consecutive temperature cycles. Gas permeance (**b**) and $H_2/CH_4$ separation factor (**c**) from the membrane M2 as a function of transmembrane pressure difference at 100 °C

changes in $E_{act-app}$ are complex to interpret because of relative changes in $E_{act}$ (higher activation energy due to the pore-shrinkage) and $\Delta E_{sur}$ (increase in binding energy with functionalized pores), a 20-fold decrease in $C_oA_{act}A_{sur}$ for $CH_4$ ($5.7 \times 10^{-7}$ to $2.8 \times 10^{-8}$, Supplementary Table 5) as a result of functionalization indicates reduced pore-density for the diffusion of $CH_4$. We envision the functionalized pore-edges would shrink in size, providing higher resistance to $CH_4$ for the diffusion, and therefore resulting in a higher gas selectivity[25]. In contrast, functionalization at 100 °C led to an increase in gas permeance by 3-fold, while the gas selectivity change slightly (Fig. 7c, d). Here, $E_{act-app}$ did not change significantly after functionalization (Supplementary Table 6), while $C_oA_{act}A_{sur}$ for gases increased by an order magnitude (Supplementary Table 7) indicating an increase in the pore-density. Given that the high-temperature functionalization leads to a higher coverage of the C–O and C=O groups, it is highly likely that these functional groups formed new pores as indicated by the HRTEM analysis.

We constructed a separation performance trajectory (Fig. 8), comparing the separation selectivity and hydrogen permeance before and after the ozone functionalization (Fig. 8). The overall trajectory trends clearly show that the gas separation performance of graphene membranes can be tuned by the ozone functionalization. A higher gas permeance (up to 300%) was achieved by generating new nanopore by ozone functionalization at 80–100 °C (membranes M7 and M8, Supplementary Figure 8 and Fig. 7c, d). A higher separation selectivity (up to 150%) was achieved by functionalization at 25 °C (M2, Fig. 7a, b). In one case, an increase in permeance, as well as separation selectivity, was obtained after ozone treatment at 80 °C for 1 min (M5, Supplementary Figure 7). Therefore, one can use ozone functionalization as a post-synthetic performance tuning method to enhance the separation performance of nanoporous single-layer graphene membranes.

Overall, we developed a scalable NPC film-assisted transfer method to fabricate crack-free and tear-free, millimeter-scale suspended single-layer CVD graphene films, allowing us to observe and understand the temperature-dependent single-component and mixture gas transport through the intrinsic defects in graphene. Graphene films with a minuscule porosity of 0.025% displayed attractive $H_2$ permeance and $H_2/CH_4$ selectivities approaching the performance of 1-μm-thick state-of-the-art polymer membranes. Improvements in the $H_2$ permeance and/or $H_2/CH_4$ selectivity were demonstrated by ozone functionalization. Overall, the methods developed here bring deployment of the single-layer nanoporous graphene membranes for gas separation a step closer to reality.

## Methods

**Graphene growth.** Single-layer graphene was synthesized by the low-pressure CVD (LPCVD) on a Cu foil (25 μm, 99.999% purity, Alfa-Aesar). Before CVD, the foil was placed in the fused quartz tube and annealed at 1000 °C in a $CO_2$ atmosphere at 700 Torr for 30 min to remove organic contaminents[49]. Then $CO_2$ was switched off and the chamber was evacuated. Following this, 8 sccm of $H_2$ was introduced to purge out $CO_2$ and to subsequently anneal the copper surface at 1000 °C. To initiate graphene nucleation, 24 sccm of $CH_4$ was added at total pressure of 460 mTorr. After 30 min growth, the $CH_4$ flow was switched off and the chamber was rapidly cooled down to the room temperature.

**Nanoporous-carbon-assisted graphene transfer.** To deposit the nanoporous carbon (NPC) film on graphene, 0.1 g block-copolymer (poly (styrene-b-4-vinyl pyridine), Polymer Source) and 0.2 g turanose (Sigma-Aldrich) were dissolved in DMF (Sigma-Aldrich). After a heat treatment of the solution at 180 °C, the solution was spin-coated onto the as-synthesized CVD graphene supported on the Cu foil. Pyrolysis of the polymer film was conducted at 500 °C in a $H_2$/Ar atmosphere for 1 h, forming the NPC film on top of graphene. The NPC/graphene/Cu was floated on a $Na_2S_2O_8$ bath (0.2 M in water) to etch the Cu foil. After Cu etching, the floating NPC/graphene film was rinsed in deionized water to remove the residues. Finally, NPC/graphene was scooped on the porous tungsten support.

functionalization was carried out at 25 °C for 2 min, the $H_2$ permeance decreased from $2.3 \times 10^{-7}$ to $1.2 \times 10^{-7}$ mol m$^{-2}$ s$^{-1}$ Pa$^{-1}$, while the $H_2/CH_4$ and the $H_2/CO_2$ selectivities increased from 10.0 to 15.0 and 5.1 to 6.4, respectively, at 150 °C (M2, Fig. 7a, b), indicating pore-shrinkage. Interestingly, both $E_{act-app}$ (defined as $E_{act} + \Delta E_{sur}$) and $C_oA_{act}A_{sur}$ decreased after the functionalization (Supplementary Table 4 and Supplementary Note 3). While the

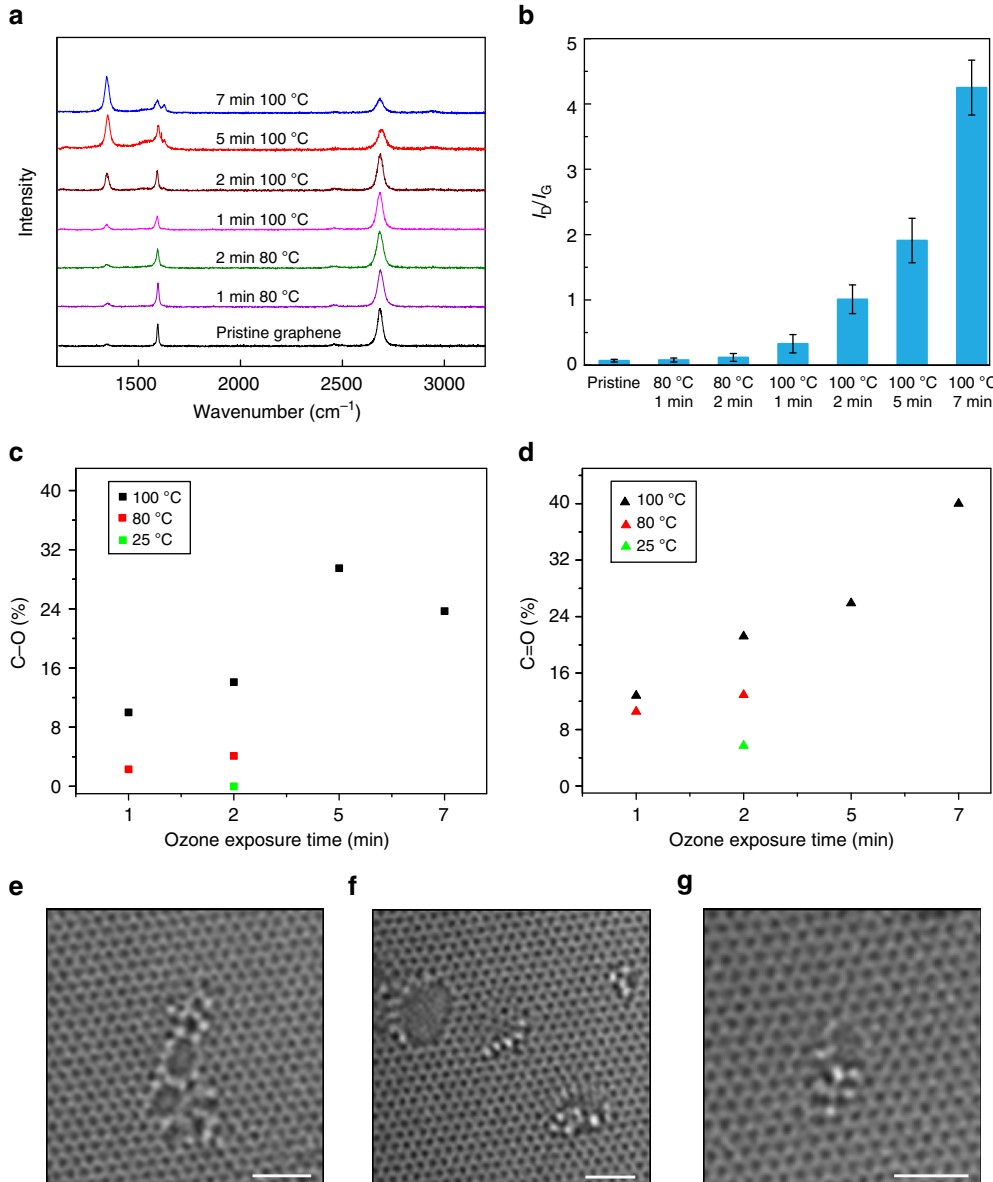

**Fig. 6** Characterization of the ozone-treated graphene. **a** Raman spectra of functionalized graphene under different functionalization conditions. **b** $I_D/I_G$ for various ozone treatment. **c**, **d** C–O (**c**) and C=O (**d**) content of the graphene as a function of the functionalization time and temperature. **e**–**g** High-resolution transmission electron microscopy (HRTEM) images of the nanopores in the ozone functionalized graphene (2 min at 80 °C). The scale bar is 1 nm. The unprocessed raw images are shown in Supplementary Figure 3d-e

**In-situ ozone functionalization**. Ozone functionalization on the suspended graphene film was conducted in-situ in the membrane module. The gas permeation module was leak-proof based on a metal–metal (Swagelok VCR fittings) seal. The membrane was sandwiched as a gasket in the VCR-based module, making a leak-tight fitting. Before, ozone functionalization, the membrane was heated to 150 °C to remove adsorbed atmospheric contaminations and to allow measurement of the gas separation performance. Then, the membrane was cooled to the functionalization temperature. Subsequently, a mixture of $O_2$ and $O_3$ (21% in $O_3$) generated by the ozone generator (Absolute Ozone® Atlas 30) was exposed to the permeate side of graphene. After a certain time, argon was used to sweep-out the residual ozone.

**Gas permeation test**. The single-component and mixture gas permeation tests were carried out in a homemade permeation cell. Permeation tests were conducted in the open-end mode.

All equipment used in the permeation setup (the mass flow controllers (MFCs) and MS) were calibrated within 5% error. The gas permeation module based on the metal–metal (Swagelok VCR fittings) seal was leak-proof. The porous tungsten support was sandwiched as a gasket in the VCR-based module, making a leak-tight fitting. To ensure temperature uniformity and accuracy, the feed and the sweep lines were preheated, and the membrane module was heated inside an oven with the temperature accuracy of ±1 °C.

A pre-calibrated MFC regulated the flow rate of feed gas, and the feed pressure was controlled by adjusting the back-pressure regulator installed at the downstream. Another pre-calibrated MFC controlled the flow rate of sweep gas (Ar), which carried the permeate gas to the pre-calibrated MS for real-time analysis of the permeate concentration. The MS was capable to read an extremely low concentration in the permeate stream. To reduce errors, MS was calibrated at low concentration of $H_2$, He, $CO_2$, and $CH_4$ in Ar, similar to those in the permeate stream.

The transmembrane pressure difference was varied between 1.5 to 7.0 bar. Before testing, all membranes were heated to 150 °C to remove the contaminations on the graphene surface. For the mixture permeation tests, an equimolar gas mixture was used on the feed side. The gas flux was calculated once the steady-state was established (typically 30 min after changing the permeation conditions). The measurements were carried out at continuously, in real-time, and only the steady-state data were reported.

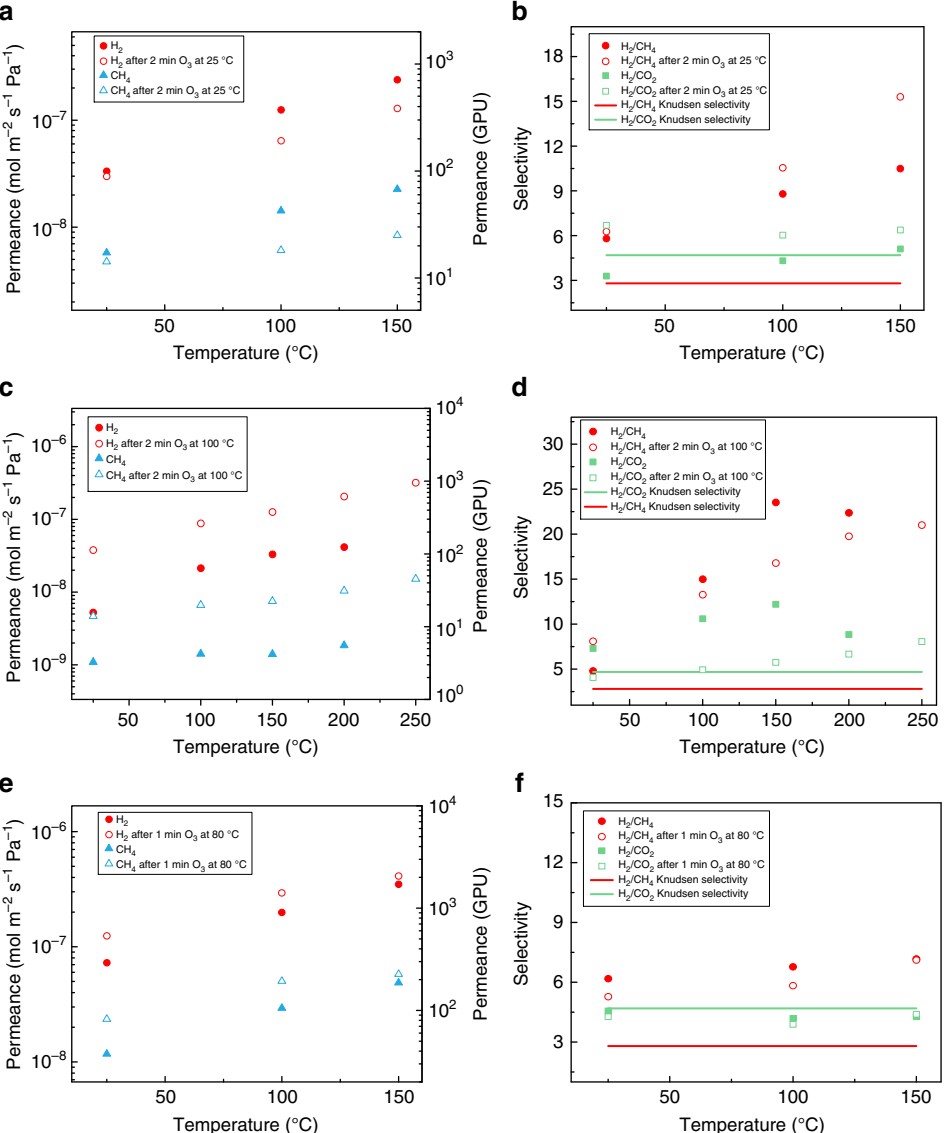

**Fig. 7** Gas separation performance of ozone-treated graphene membranes. **a**, **b** Gas separation performance of M2 treated by 2 min $O_3$ at 25 °C, **a** gas permeance of $H_2$ and $CH_4$, **b** gas selectivity of $H_2/CH_4$ and $H_2/CO_2$. **c**, **d** Gas separation performance of M8 treated by 2 min $O_3$ at 100 °C, **c** gas permeance of $H_2$ and $CH_4$, **d** gas selectivity of $H_2/CH_4$, and $H_2/CO_2$. **e**, **f** Gas separation performance of M6 treated by 1 min $O_3$ at 80 °C, **e** gas permeance of $H_2$ and $CH_4$, **f** gas selectivity of $H_2/CH_4$ and $H_2/CO_2$

**Electron microscopy**. SEM was carried out by using FEI Teneo SEM at 0.8–2.0 kV and working distances of 2.5–9.0 mm. No conductive coating was applied on the substrates prior to SEM. TEM imaging and electron diffraction of the NPC film and the composite graphene/NPC film were conducted by FEI Tecnai G2 Spirit Twin with 120 keV incident electron beam.

For HRTEM, graphene was transferred on a quantifoil TEM grid by the traditional wet-transfer technique[28]. Briefly, a thin poly(methyl methacrylate) or PMMA film was spin-coated on top of graphene. Following this, Cu was etched in a sodium persulfate bath. After rinsing the floating graphene/PMMA film with deionized water, the composite film was transferred to the TEM grid. Subsequently, PMMA was removed by heating the sample to 400 °C in a reducing atmosphere of $H_2/Ar$.

Aberration-corrected (Cs) HRTEM was performed using a double-corrected Titan Themis 60-300 (FEI) equipped with a Wein-type monochromator. To reduce the electron radiation damage, an 80 keV incident electron beam was used for all experiments. The incident electron beam was monochromated ("rainbow" mode illumination) to reduce the effects of chromatic aberration, and a negative Cs of ~15–20 μm and slight over focus were used to give a "bright atom" contrast in the images. HRTEM images were post-treated using a combination of Bandpass and Gaussian filters to reduce noise and improve contrast.

**Raman spectroscopy**. Raman characterization was carried on graphene transferred onto the $SiO_2$/Si wafer by the wet-transfer method[28]. Single-point data collection and mapping were performed using Renishaw micro-Raman spectroscope (532 nm, 2.33 eV, ×100 objective). Analysis of the Raman data was carried out using MATLAB. For calculation of the D and the G peak height, the background was subtracted from the Raman data using the least-squares curve fitting tool (lsqnonlin).

**X-ray photoelectron spectroscopy**. The X-ray photoelectron spectroscopy (XPS) analysis was conducted on the graphene mounted on Cu foil using a Mg Kα X-ray source (1253.6 eV) and Phoibos 100 (SPECS) hemispherical electron analyser with multichanneltron detector. The XPS spectra were recorded in fixed analyser transmission (FAT) mode using pass energies of 90 eV for the survey and 20 eV for the narrow scans. The samples did not show electrostatic charging thus the binding energies are presented without any correction (Bonding energy of C–C: 284.4 eV; C–O: 285.7 eV; C=O: 286.8 eV; O–C=O: 288.5 eV). Because carbonyl group (C=O) is part of (O–C=O), O–C=O was counted in C=O in the summation of functional group. The XPS spectra were processed with CasaXPS, with background subtraction by the Shirley method.

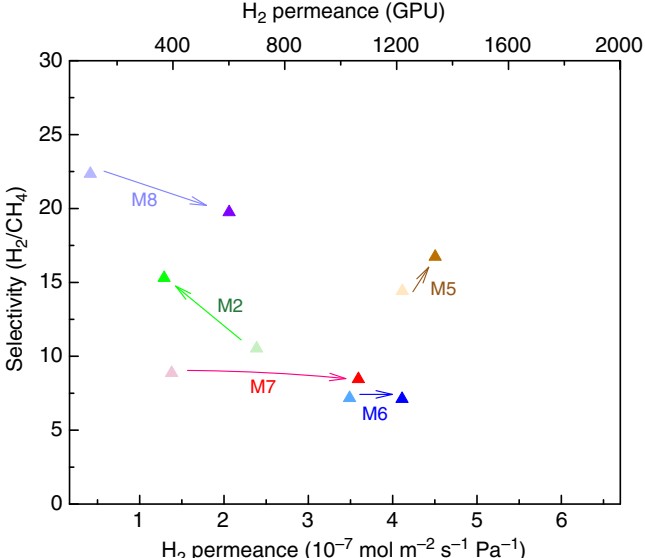

**Fig. 8** The evolution of gas separation performance after different ozone treatments. Permeance data for membrane M8 is at 200 °C, and all other (M2, M5, M6, and M7) is at 150 °C. Light and dark markers represent the gas performance from the intrinsic defects and the ozone-treated graphene, respectively

**Data availability**. The authors declare that all the data supporting the findings of this study are available within the article (and its Supplementary Information file), or available from the corresponding author on reasonable request.

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

## Acknowledgements

We acknowledge our home institution (EPFL), GAZNAT, Swiss Competence Center of Energy Research – Efficiency in Industrial Processes (SCCER-EIP), and the ETH Board for funding the project.

## Author contributions

K.A. and S.H. conceived the project. S.H. designed and fabricated the graphene membranes. M.D. and S.H. developed the NPC film. W.L. and A.Z. characterized graphene by XPS. E.O. and D.T.L.A. characterized graphene with HRTEM. K.A., S.H., M.D., G.H., M. R., and J.Z. built the CVD and the membrane setup. K.A. and M.S.S. discussed the membrane module design. S.H. and K.A. wrote the paper. All authors revised the paper.

## Additional information

**Competing interests:** K.A, S.H., M.D., and G.H. hold filed patent applications related to the developments on the NPC film and the graphene transfer. The remaining authors declare no competing interests.

