## [Peer Review File · Nature Communications]

Reviewer's comments

Reviewer #1 (Remarks to the Author):

In this work, the authors describe results from a new graphene transfer method using nanoporous graphene. One of the significant challenges of moving graphene from small scale proof-of-principle experiments to larger scale, applications-driven studies has been the difficulty in transferring graphene without creating unwanted tears and defects. So the manuscript presents very nice results that would be of broad interest to the scientific community. With that in mind, I think the manuscript is suitable for publication. I only have a couple of minor points to raise before publication. There is a wide variability of the single component permeance values (Fig. 3a, b). I assume this is due to the variability of the number of intrinsic defects in the as synthesized graphene. Maybe the authors make that point but I didn't see it so just a sentence explaining that variability would be useful. Along those same lines, the effect of the ozone etching also shows considerable variability. For example, samples M5 and M6 both were exposed at 80 degC for 1 min with somewhat different results. M2 was exposed at lower temperature but longer time and showed a decrease in selectivity. Why would the pore shrink after ozone etching? These results show that there remains some uncertainty in tuning the porosity of the graphene. Despite that, the manuscript shows excellent progress in this area.

Reviewer #2 (Remarks to the Author):

The authors have demonstrated crack-free transfer of large-area single-layer graphene membranes for selective gas separation. They used a smart idea to protect the single-layer graphene, by coating it with a thin film of nanoporous carbon. This novel approach apparently made the single-layer graphene dramatically more robust. Then the authors leveraged the intrinsic porosity in the single-layer graphene for molecular sieving of gases. They demonstrated good selectivities and permeance for H₂/CH₄ separation, despite the low porosity of the graphene membrane. They further used chemical functionalization to show improved permeance. This is a great advance upon the previous work in demonstrating the potential of porous graphene as the ultimate membrane for gas separation. I enthusiastically support the publication of this work, with the following minor comments.

1. Given the low porosity, the measured permeances of 10^2 to 10^3 GPU are quite impressive. Still, molecular dynamics simulations have shown that 10^5 GPU can be achievable. Maybe the authors can comment on the gap between their measurements and the simulations, so that one knows there is more space for the experiments to chase even higher permeances.
2. Due to the size difference between H₂ and CH₄, one would expect a higher H₂/CH₄ selectivity. Presumably this is due to the pore size dispersion in the graphene layer. Can the authors comment on the pore size dispersion in their graphene layer?
3. I wonder if the technique can be used to generate bilayer graphene membranes that can still show gas separation capability.

Reviewer #3 (Remarks to the Author):

The authors report on a method of preparing single layer graphene membranes. The prepared membranes were characterised by a number of experimental materials techniques. This characterisation work supports the major claim of a novel process to prepare single layer graphene membranes.

In addition, the authors carried out a large number of gas permeation tests to determine the performance of the graphene membranes. The gas permeation results are credible and within the range of gas

permeance reported in literature for microporous inorganic membranes. Overall the reported results are modest.

Graphene has recently attracted the concerted effort of the research community around the world. In particular, graphene membrane publications have increased significantly in the last 3-4 years, though generally for liquid processing. Graphene membranes for gas separation have proved to be difficult due to low performance, and the number of papers reported in literature are limited.

This work will be attractive for those researching in this growing field of graphene membranes, and warrants publication in NatComm. However, there are several areas of this paper that require further clarification, or proof, for the benefit of the readers of NatComm. Therefore, a major review is recommended as set out below.

1) Title: the words “large-area” used in the title are far-fetched. Currently, graphene membranes produced by other methods are at least two orders of magnitude higher than the 1 mm^2 reported in this work. The authors may mean a relatively large-area as compared with previous work in the field. Authors should re-consider the title of their work.

2) Authors have shown SEM/TEM images of the graphene membranes only. To strengthen the claim of a single layer graphene membrane, authors should also show SEM/TEM images of the cross section showing the graphene single layer on the top of the substrate.

3) Experimental errors for gas permeation tests were not provided. Generally it is expected experimental errors in the order of $\pm 10\%$ based on calibration of mass flow controllers, calibration of GC, variation in the temperature of the furnace, area measurements etc. This is particularly important as the membrane area tested was very small. So the measured gas flow rates using argon as a sweep gas could be very low based on the permeance results. The error here could be significant. Authors should make an effort to show experimental variation which is consistent with reporting results in any competent journal in the field.

4) It is interesting that authors are comparing their membrane performance results against dense polymeric membranes from Robinson plot. This type of comparison is not appropriate as polymers are dense membranes where the transport phenomena differs from graphene membranes. Graphene is classified as an inorganic material, and in the case of this work the membrane is porous and not dense. To provide a more convincing comparison, authors should compare against inorganic porous membranes for gas separation such as zeolite and silica membranes as examples.

5) Initially this reviewer was confused with the terms used in Eq. 1. This equation for activated transport is in fact the same equation developed by Barrer for the transport of gases in porous material, which was further developed and adopted by Burggraaf and other researchers in the 1990s. The problem here is the use of the term translocation. It is not clear if this a surface diffusion term, or a micropore diffusion term. This confusion raised the question of how this term was calculated. By reading the supplementary information, it then became clear to me that first the authors calculated the apparent energy of activation from the measured fluxes at various temperatures, thus using an Arrhenius relationship. So in principle, the term used here as energy of activation for translocation is the term conventionally used by those working in the area of microporous membranes as the Energy of diffusion through micropores. The energy term is the energy of adsorption. The authors do not mention this term as energy of adsorption, but mention in the same page (4) competitive adsorption. The authors should clarify the use of these terms to be more in line with the terms conventionally used and understood by those working in the field of microporous. Also using the term E_{act} for the diffusion may be confusing too.

6) As equation 1 is basically the same as model pioneered by Barrer and then adopted by Burggraaf and co-workers for microporous diffusion, authors should give credit to them by citing their papers.

7) The calculation of the energy of diffusion through micropores is based on the apparent energy of activation and the energy of surface adsorption. It is surprising that helium has an energy of surface adsorption based on results given above Table S2. Minor helium adsorption has been proven for temperatures close to zero Kelvin degrees. Under the testing conditions in this work, helium adsorption should be non-measurable and zero. This reviewer wonders where the proof that helium is adsorbing in graphene at the testing temperatures in this work is.

8) In Fig. 3 authors show the permeance for hydrogen only, and chose to show gas pair selectivity to show the remainder of the results. This is unusual as in microporous inorganic membranes the permeance results tend to be shown for all gases tested. This comes to the point of Fig. 3c for H₂/CO₂ separation. The reduction in selectivity at 200C suggest that has been a change in the permeance of CO₂. This reviewer wonders as to whether CO₂ permeance has decreased with temperature from room temperature testing, or just at 200C?

9) The use of a few hyperbolic adjectives needs moderation. For instance:

(i) Abstract: yielding ultrahigh permeance. In fact the permeances are moderate as there are reports for micropore inorganic membranes reaching permeances of one order of magnitude higher than those in this work.

(ii) Last paragraph page 4: authors claim that the graphene membranes showed exceptional thermal stability. This claim of “exceptional” is inappropriate as it is based on a short experimental work, possibly in a single day for each membrane. There are reports in literature showing inorganic membranes tested continuously for 2000 h at high temperatures. Generally in the area of membranes 1000 h testing would be considered as proof of exceptional thermal stability.

10) Page 4 top paragraph: Please change “absorbed” phase transport to “adsorbed”.

11) Section S3: Please add more information on the experimental details for gas permeation. For instance, state if dead-end mode was used for single gas permeation. In the case of gas mixture, was the valve at the retentate line left closed (dead-end) or open (open-end)?

12) Finally, the membrane work preparation in this paper is quite complex. Hence, scaling up to industrial application may be difficult. However, sensors generally require small areas and could be a potential application of the method developed in this paper.

Response to the reviewer's comments

Reviewer 1:

In this work, the authors describe results from a new graphene transfer method using nanoporous graphene. One of the significant challenges of moving graphene from small scale proof-of-principle experiments to larger scale, applications-driven studies has been the difficulty in transferring graphene without creating unwanted tears and defects. So the manuscript presents very nice results that would be of broad interest to the scientific community. With that in mind, I think the manuscript is suitable for publication. I only have a couple of minor points to raise before publication. There is a wide variability of the single component permeance values (Fig. 3a, b). I assume this is due to the variability of the number of intrinsic defects in the as synthesized graphene. Maybe the authors make that point but I didn't see it so just a sentence explaining that variability would be useful. Along those same lines, the effect of the ozone etching also shows considerable variability. For example, samples M5 and M6 both were exposed at 80 deg C for 1 min with somewhat different results. M2 was exposed at lower temperature but longer time and showed a decrease in selectivity. Why would the pore shrink after ozone etching? These results show that there remains some uncertainty in tuning the porosity of the graphene. Despite that, the manuscript shows excellent progress in this area.

Author's response: We thank the reviewer for supporting the publication. The reviewer correctly noted that the variability in the transport from the intrinsic defects is due to variability in the number density of the intrinsic defects in the as-synthesized graphene. The density of intrinsic defects in CVD graphene can vary from batch to batch, creating a variability in the gas permeance. However, we note that the nature of hydrogen transport did not change across the membranes indicating that the pore-size distribution (PSD) did not change significantly between the membranes. All eight membranes displayed the activated transport of hydrogen with an average activation energy of 20.2 ± 2.7 kJ mol⁻¹. As per the reviewer's recommendations, we have added a sentence on the page 4 of the manuscript,

"We note that while the gas permeance varied across the eight membranes, the activation energy for pore-translocation was consistent across the membranes. This indicates that while the PSD was uniform across the membranes, the density of the intrinsic defects varied across the membranes."

We agree with the reviewer that there is an added variability in the ozone functionalization results mainly because chemical functionalization and etching is extremely sensitive to even the slight variations in the PSD of the intrinsic defects. This is mainly because the reactivity of carbon at the graphene pore-edge is orders of magnitude faster than that of carbon in the basal plane of graphene.

The reviewer has perhaps incorrectly interpreted the results for the membrane M2. The H₂/CH₄ selectivity increased for M2 (not decreased as the reviewer noted) as a result of functionalization at the lower temperature, indicating pore-shrinkage. The pore-shrinkage is expected because the pore-edges are likely to be functionalized with epoxy and carbonyl group post-ozone functionalization, reducing the electron-density-gap in the nanopore.

Reviewer 2:

The authors have demonstrated crack-free transfer of large-area single-layer graphene membranes for selective gas separation. They used a smart idea to protect the single-layer graphene, by coating it with a thin film of nanoporous carbon. This novel approach apparently made the single-layer graphene dramatically more robust. Then the authors leveraged the intrinsic porosity in the single-layer graphene for molecular sieving of gases. They demonstrated good selectivities and permeance for H₂/CH₄ separation, despite the low porosity of the graphene membrane. They further used chemical functionalization to show improved permeance. This is a great advance upon the previous work in demonstrating the potential of porous graphene as the ultimate membrane for gas separation. I enthusiastically support the publication of this work, with the following minor comments.

Author's response: We thank the reviewer for supporting the publication.

1. Given the low porosity, the measured permeances of 10² to 10³ GPU are quite impressive. Still, molecular dynamics simulations have shown that 10⁵ GPU can be achievable. Maybe the authors can comment on the gap between their measurements and the simulations, so that one knows there is more space for the experiments to chase even higher permeances.

Author's response: We thank the reviewer for the helpful comments. We have modified a statement in the manuscript on page 5 to highlight that higher permeances can indeed be achieved.

“Theoretically, the H₂ permeance can be further increased beyond 10⁵ GPU by increasing the defect-density to 10¹² – 10¹³ cm⁻².”

2. Due to the size difference between H₂ and CH₄, one would expect a higher H₂/CH₄ selectivity. Presumably this is due to the pore size dispersion in the graphene layer. Can the authors comment on the pore size dispersion in their graphene layer?

Author's response: We agree with the reviewer that it is the pore-size-distribution (PSD) of intrinsic defects in the CVD graphene limits the H₂/CH₄ selectivity to less than 25. The manuscript had a statement on page 3, “The H₂/CH₄ selectivity was lower than that from Bi-3.4 membrane²⁰, indicating a wider PSD of intrinsic defects in CVD graphene, compared to PSD from pores incorporated in micromechanically exfoliated graphene.” A low-density of intrinsic defects (0.025%) makes it difficult to survey a large number of lattice-resolved HRTEM images for determining the PSD.

Estimating the number density of larger pores, we have added a section (at end of the section S4) in the Supplementary Information including a Table (Table S8) listing the percentage of defects with an electron density gap larger than the size of CH₄ (kinetic diameter of 0.38 nm). Given that the transmission coefficient from such pores (effusive transport) is ca. 4 orders of magnitude higher than that from the smaller pores (activated transport), we show that the percentage of pores hosting an electron-density-gap larger than 0.38 nm is very small (less than 25 ppm).

Table S8. Estimated density of large, non-selective pores in graphene as a function of α_{H_2/CH_4}

α_{H_2/CH_4}	C_e/C_a	PPM of nanopores (with respect to all nanopores) with an electron density gap larger than 0.38 nm
5	2.5×10^{-5}	25.0
10	1.1×10^{-5}	11.1
15	7.1×10^{-6}	7.1

20	5.3×10^{-6}	5.3
25	4.2×10^{-6}	4.2

We have also added a statement in the main manuscript on page 3.

“Based on the achieved H₂/CH₄ selectivities, the estimated percentage of larger nanopores yielding non-selective effusive gas transport is less than 25 ppm (refer to the section S4 and Table S8 for more details).”

3. I wonder if the technique can be used to generate bilayer graphene membranes that can still show gas separation capability.

Author’s response: The transfer technique introduced here can be adapted to make bilayer and multilayer graphene membranes, using approaches similar to that in Celebi et al., Science, 344, 289, 2014.

Reviewer 3

The authors report on a method of preparing single layer graphene membranes. The prepared membranes were characterised by a number of experimental materials techniques. This characterisation work supports the major claim of a novel process to prepare single layer graphene membranes.

In addition, the authors carried out a large number of gas permeation tests to determine the performance of the graphene membranes. The gas permeation results are credible and within the range of gas permeance reported in literature for microporous inorganic membranes. Overall the reported results are modest.

Graphene has recently attracted the concerted effort of the research community around the world. In particular, graphene membrane publications have increased significantly in the last 3-4 years, though generally for liquid processing. Graphene membranes for gas separation have proved to be difficult due to low performance, and the number of papers reported in literature are limited.

This work will be attractive for those researching in this growing field of graphene membranes, and warrants publication in NatComm.

Author's response: We thank the reviewer for supporting publication in Nature Communications.

However, there are several areas of this paper that require further clarification, or proof, for the benefit of the readers of NatComm. Therefore, a major review is recommended as set out below.

1) Title: the words “large-area” used in the title are far-fetched. Currently, graphene membranes produced by other methods are at least two orders of magnitude higher than the 1 mm² reported in this work. The authors may mean a relatively large-area as compared with previous work in the field. Authors should re-consider the title of their work.

Author's response: To the best of our knowledge, crack-free single-layer graphene membrane for molecular separation has been limited to 5 μm in diameter¹⁻³. Compared with those, 1 mm² is a large-area. By, “graphene membranes produced by other methods”, we believe that the reviewer is referring to multilayered/stacked graphene-oxide nanosheets based membranes. If this is the case, we will like to note that these two membrane technologies have very different performance capabilities. Single-layer graphene can much reach higher gas permeance upon incorporation of a moderate density of nanopores.

2) Authors have shown SEM/TEM images of the graphene membranes only. To strengthen the claim of a single layer graphene membrane, authors should also show SEM/TEM images of the cross section showing the graphene single layer on the top of the substrate.

Author's response: In the submitted manuscript, we included micro-Raman spectroscopy based analysis of CVD graphene (Fig. S4). The histogram of I_{2D}/I_G conclusively indicated that the graphene being studied here is single-layer (2 < I_{2D}/I_G < 3). Moreover, the electron diffraction obtained from the suspended composite film (nanoporous carbon/graphene, Fig. 2d) also conclusively points to single-layer graphene.

3) Experimental errors for gas permeation tests were not provided. Generally it is expected experimental errors in the order of +/- 10% based on calibration of mass flow controllers, calibration of GC, variation in the temperature of the furnace, area measurements etc. This is particularly important as the membrane area tested was very small. So the measured gas flow rates using argon as a sweep gas could be very low based on the permeance results. The error here could be significant. Authors should make an effort to show experimental variation which is consistent with reporting results in any competent journal in the field.

Author's response: We thank the reviewer for highlighting the importance of experimental error. All the equipment used in the permeation setup (mass flow controller and mass spectrometer) were calibrated within 5% error, which is acceptable in an experimental study. The gas permeation module, based on the metal-metal (Swagelok VCR fittings) seal was leak-proof. Here, the W support was sandwiched as a gasket in the VCR fitting, making a leak-tight fitting. To ensure uniformity in the temperature, the feed and the sweep gas lines and the membrane module were heated inside an oven with the temperature accuracy of ± 1 °C. The feed and sweep lines were pre-heated to ensure temperature uniformity.

We used a mass spectrometer and not GC (please refer to section S3), which was calibrated at the low concentrations of H₂, He, CO₂ and CH₄ in Ar flow, similar to those measured in the permeate side of the membrane. This eliminated calibration related errors, and certainly any significant error that the reviewer is referring to. The measurements were carried out at continuously, in real-time, and the steady state data was reported.

For the interest of readers, we have added more details in the section S3 of the Supplementary Information (see below):

“The single-component and mixture gas permeation tests were carried out in a homemade permeation cell. Permeation tests were conducted in the open-end mode.

All equipment used in the permeation setup (the mass flow controllers (MFCs) and mass spectrometer (MS)) were calibrated within 5% error. The gas permeation module, based on the metal-metal (Swagelok VCR fittings) seal was leak-proof. The porous W support was sandwiched as a gasket in the VCR based module, making a leak-tight fitting. To ensure temperature uniformity and accuracy, the feed and the sweep lines were preheated, and the membrane module was heated inside an oven with the temperature accuracy of ± 1 °C.

A pre-calibrated MFC regulated the flow rate of feed gas, and the feed pressure was controlled by adjusting the back-pressure regulator installed at the downstream. Another pre-calibrated MFC controlled the flow rate of sweep gas (Ar), which carried the permeate gas to the pre-calibrated MS for real-time analysis of the permeate concentration. The MS was capable to read an extremely low concentration in the permeate stream. To reduce errors, MS was calibrated at low concentration of H₂, He, CO₂ and CH₄ in Ar, similar to those in the permeate stream.

The transmembrane pressure difference was varied between 1.5 to 7.0 bar. Before testing, all membranes were heated to 150 °C to remove the contaminations on the graphene surface. For the mixture permeation tests, an equimolar gas mixture was used on the feed side. The gas flux was calculated once the steady-state was established (typically 30 minutes after changing the permeation conditions). The measurements were carried out at continuously, in real-time, and only the steady-state data were reported.”

4) It is interesting that authors are comparing their membrane performance results against dense polymeric membranes from Robinson plot. This type of comparison is not appropriate as polymers are dense membranes where the transport phenomena differs from graphene membranes. Graphene is classified as an inorganic material, and in the case of this work the membrane is porous and not dense. To provide a more convincing comparison, authors should compare against inorganic porous membranes for gas separation such as zeolite and silica membranes as examples.

Author's response: We thank the reviewer for the helpful comments. Based on the reviewer's recommendation, we have added the comparison with inorganic nanoporous membranes in Fig. S6. The polymeric membranes for H₂/CH₄ separation is most widely used membranes, and are therefore also included in the comparison.

We note that with only 0.025% porosity in single-layer graphene, the H₂ permeance of the single-layer graphene falls short of that from the zeolites and MOF membranes which have been developed over several decades. This is the first report of gas-selective single-layer graphene membrane, and we are optimistic that future developments on increasing the pore-density in graphene will lead to a superior performance, as predicted by the molecular simulations.

Fig. S6. A comparison of single-layer graphene films (all eight membranes) with a low-density of intrinsic defects (0.025%) with membranes in the literature in terms of the H₂/CH₄ separation performances. The red line is the polymer upper bound assuming 1 μm -thick skin layer¹³, ZIF-8¹⁴, silica^{15,16} and zeolite membranes (SSZ-13¹⁷, AIPO-18¹⁸, DDR¹⁹, LTA²⁰, CHA²¹).

5) Initially this reviewer was confused with the terms used in Eq. 1. This equation for activated transport is in fact the same equation developed by Barrer for the transport of gases in porous material, which was further developed and adopted by Burggraaf and other researchers in the 1990s. The problem here is the use of the term translocation. It is not clear if this a surface diffusion term, or a micropore diffusion term. This confusion raised the question of how this term was calculated. By reading the supplementary information, it then became clear to me that first the authors calculated the apparent energy of activation from the measured fluxes at various temperatures, thus using an Arrhenius relationship. So in principle, the term used here as energy of activation for translocation is the term conventionally used by those working in the area of microporous membranes as the Energy of diffusion through micropores. The energy term is the energy of adsorption. The authors do not mention this term as energy of adsorption, but mention in the same page (4) competitive adsorption. The authors should clarify the use of these terms to be more in line with the terms conventionally used and understood by those working in the field of microporous. Also using the term E_{act} for the diffusion may be confusing too.

Author's response: We thank the reviewer for the helpful comments. To make sure that the text is easy to understand, we have edited the manuscript page 4 as following,

“Here, C_0 is the pore-density, E_{act} is the activation energy for the pore-translocation event, and ΔE_{sur} is the adsorption energy of gas on to the graphene nanopore. A_{act} is the pre-exponential factor for the pore-translocation. A_{sur} is the pre-exponential factor for the adsorption event, representing changes in the overall entropy.”

6) As equation 1 is basically the same as model pioneered by Barrer and then adopted by Burggraaf and co-workers for microporous diffusion, authors should give credit to them by citing their papers.

Author's response: We thank the reviewer for the helpful comments. We have now cited the work of Barrer and Burggraaf on page 4. The sentence now reads as

To understand the transport behavior, the activation energy for translocation of gases was extracted from the temperature-dependent gas flux using an adsorbed phase transport model developed for nanoporous graphene^{36,37} using the concepts of adsorption and diffusion^{38,39}.

7) The calculation of the energy of diffusion through micropores is based on the apparent energy of activation and the energy of surface adsorption. It is surprising that helium has an energy of surface adsorption based on results given above Table S2. Minor helium adsorption has been proven for temperatures close to zero Kelvin degrees. Under the testing conditions in this work, helium adsorption should be non-measurable and zero. This reviewer wonders where the proof that helium is adsorbing in graphene at the testing temperatures in this work is.

Author's response: We obtained the adsorption energy of He (2 kJ/moles) based on the reported value on graphite^{4,5}, and our own calculations of the adsorption energy on the surface of graphene. The literature^{4,5} points out that the He adsorption energy is close to 1.5 kJ/moles. These are cited in the Supplementary Information, and also list them here.

Hellemans R, Van Itterbeek A, Van Dael W. The adsorption of helium, argon and nitrogen on graphite. Physica, 1967, 34(3): 429-437.

Cole M W, Frankl D R, Goodstein D L. Probing the helium-graphite interaction. Reviews of Modern Physics, 1981, 53(2): 199.

Our Lennard-Jones based interaction calculation of gases on graphene⁶ points to an adsorption energy of 2 kJ/mol for helium.

We also note that a small change in adsorption energy (1-2 kJ/mole) will not significantly change the extracted activation energies and the derived conclusions in the manuscript.

8) In Fig. 3 authors show the permeance for hydrogen only, and chose to show gas pair selectivity to show the remainder of the results. This is unusual as in microporous inorganic membranes the permeance results tend to be shown for all gases tested. This comes to the point of Fig. 3c for H₂/CO₂ separation. The reduction in selectivity at 200C suggest that has been a change in the permeance of CO₂. This reviewer wonders as to whether CO₂ permeance has decreased with temperature from room temperature testing, or just at 200 C?

Author's response: The CO₂ transport was activated, and as a result, CO₂ permeance through the graphene membrane increased with temperature.

To make it clear, we have added all the gas permeance data in the Supplementary Information section S6, Tables S9-S11.

9) The use of a few hyperbolic adjectives needs moderation. For instance:

(i) Abstract: yielding ultrahigh permeance. In fact the permeances are moderate as there are reports for micropore inorganic membranes reaching permeances of one order of magnitude higher than those in this work.

Author's response: The adjective "ultrahigh" in the abstract is used to highlight the ultimate performance potential of the single-layer nanoporous graphene membrane indicated by several molecular simulations^{7,8}.

The exact text in the abstract is as follows, "The single-layer nanoporous graphene film has the potential to be the ultimate membrane, yielding ultrahigh permeance and attractive molecular selectivity."

The H₂ permeance reported in this work is exceptionally high considering the extremely small porosity (0.025%), and confirms for the first time that upon increasing the porosity to even 1% will lead to ultrahigh permeance.

(ii) Last paragraph page 4: authors claim that the graphene membranes showed exceptional thermal stability. This claim of "exceptional" is inappropriate as it is based on a short experimental work, possibly in a single day for each membrane. There are reports in literature showing inorganic membranes tested continuously for 2000 h at high temperatures. Generally in the area of membranes 1000 h testing would be considered as proof of exceptional thermal stability.

Author's response: We have changed the sentence as following on page 5,

"The graphene membranes were thermally stable (Fig. 4a)."

10) Page 4 top paragraph: Please change "absorbed" phase transport to "adsorbed".

Author's response: We thank the reviewer for pointing out the typographical mistake. We have fixed it now.

11) Section S3: Please add more information on the experimental details for gas permeation. For instance, state if dead-end mode was used for single gas permeation. In the case of gas mixture, was the valve at the retentate line left closed (dead-end) or open (open-end)?

Author's response: We thank the reviewer for the helpful comment which we think will improve the quality of the manuscript. As mentioned in the comment 3, we have added more details to the experimental section including that the experiment was done in the open-end mode.

12) Finally, the membrane work preparation in this paper is quite complex. Hence, scaling up to industrial application may be difficult. However, sensors generally require small areas and could be a potential application of the method developed in this paper.

Author's response: We thank the reviewer for putting forth their opinion about the scale-up of graphene membrane. We will like to point reviewer to several papers on the roll-to-roll production of single-layer CVD graphene (references 22 and 23 in the main manuscript).

References

1. Koenig, S. P., Wang, L., Pellegrino, J. & Bunch, J. S. Selective molecular sieving through porous graphene. *Nat. Nanotechnol.* **7**, 728–32 (2012).
2. Agrawal, K. V. *et al.* Fabrication, Pressure Testing and Nanopore Formation of Single Layer Graphene Membranes. *J. Phys. Chem. C* **121**, 14312–14321 (2017).
3. Surwade, S. P. *et al.* Water desalination using nanoporous single-layer graphene. *Nat. Nanotechnol.* **10**, 459–464 (2015).
4. Hellemans, R., Van Itterbeek, A. & Van Dael, W. The adsorption of helium, argon and nitrogen on graphite. *Physica* **34**, 429–437 (1967).
5. Cole, M. W., Frankl, D. R. & Goodstein, D. L. Probing the helium-graphite interaction. *Rev. Mod. Phys.* **53**, 199–210 (1981).
6. Yuan, Z. *et al.* Mechanism and Prediction of Gas Permeation through Sub-Nanometer Graphene Pores: Comparison of Theory and Simulation. *ACS Nano* **11**, 7974–7987 (2017).
7. Liu, H., Dai, S. & Jiang, D. E. Permeance of H₂ through porous graphene from molecular dynamics. *Solid State Commun.* **175–176**, 101–105 (2013).
8. Lee W. Draushuk and Michael S. Strano. Mechanisms of Gas Permeation through Single Layer Graphene Membranes. *Langmuir* **28**, 16671–16678 (2012).

Reviewer's comments

Reviewer #1 (Remarks to the Author):

Editorial note: this reviewer provided comments to the editor only.

Reviewer #2 (Remarks to the Author):

The authors have addressed my previous comments.

Reviewer #3 (Remarks to the Author):

The authors carried out a revision of their manuscript where many questions were adequately addressed. However, there are several points which still require clarification as set out below:

1) This reviewer is still concerned with the concept of “large area”. This concept is relative to graphene membranes only and it is not absolute to membrane areas conventionally reported for other graphene membranes, or to any other type of membranes. This concept may give the wrong impression that extremely low areas as in the case of this paper can be considered as large area for those working in membrane science and technology.

2) Thanks and the point made is agreed.

3) Thanks and the edits have addressed this question.

4) Thanks for adding more points to figure S6 (Robinson Plot). Please note that inorganic membranes are also extensively used for industrial applications. Further, the comparison to polymeric membranes is a weak argument in terms of separation, as polymeric membranes are the lower denominator (i.e. low selectivity). As graphene membranes are inorganic membranes, a solid comparison with inorganic membranes is warranted. Further, some of the results added in figure S6 are actually not the best results, and authors are still comparing their results against mid-level membrane performance and missed several important performance results. For instance, Verweij's group published a paper in Science (de Vos & Verweij, Science 1998, 5357, 1710-1711) reporting silica membranes with H₂ permeances 5 times higher than the membranes in this work, whilst delivering H₂/CH₄ selectivities of 770. There are many other excellent results and authors should improve this aspect of their paper.

5) Thanks for adding the explanation on the activated model. Could you please change the term translocation to diffusion? Translocation is generally used for protein transport through membranes (see Verner & Schatz, Science, 1988, 241, 1307-1313). Diffusion is the correct term used for the transport of gas through membranes. There is an immense amount of literature on gas diffusion for membranes, and using this correct term will be more consistent with the understanding of those readers and researchers working in membrane science and technology for gas separation. Further, the energy of activation is consistent with the term energy of diffusion of gas molecules.

6) Thanks for citing the works from Barrer and Burggraaf. Please change “nanoporous graphene” to “microporous materials”. Please note that microporous materials are those with pore diameters below 2nm according to IUPAC, which is extensively used in studies about inorganic membranes complying with activated transport.

7) Again, the paper quoted on experimental work of He adsorption on graphene was at extremely low temperature (liquid nitrogen temperatures). The point here is that if the authors carry out a He adsorption on graphite at the temperature range used in this work (25 to 200C), He adsorption is likely not to occur. In this case, the apparent energy of activation will be almost equal to the energy of diffusion of He through the membrane.

8) This is an interesting result which was not clear before, as gas selectivities were reported. High quality inorganic membranes complying with activated transport generally deliver a negative apparent activation energy for the transport of CO₂ (i.e. flux or permeance decreases with temperature). This is mainly due to the fact that the CO₂ heat of sorption is higher than the CO₂ energy of diffusion. In the case of this work, the opposite effect is reported as CO₂ has a positive apparent activation energy. Any comments please.

9) This reviewer is still concerned with the concept of ultrahigh permeance. The permeances of the graphene membranes in this work are not high and the gas selectivities are low. This is not a criticism to the novelty of this work. However, it is a criticism to these statements which will not be accepted by those working in the membrane community. Let us compare the results in this paper with those published by Verweij's group in Science (de Vos & Verweij, Science 1998, 5357, 1710-1711). There are also a number of papers given similar performance as those published by Verweij's group. The arguments here are as follows:

- H₂ permeance were 5 times higher than this work and the silica membranes had a thickness of 30 nm.
- The permeance is inversely proportional to the thickness of the membrane. If the silica membrane could be produced with a thickness of 1 nm (like single graphene membranes), than silica membranes would have a permeance of 150 times higher than the graphene membranes in this work.
- This reviewer agrees that the porosity of the graphene membrane is low (0.025%). Silica membranes have porosities ranging from close to 0% all the way to 30%. Let us assume a high porosity value of 25% for the silica membranes. So the porosity of the graphene membranes is 1000 times lower.
- Then if we compare the permeances (150 times) over the porosity (1000 times), the graphene membrane will be a factor of 8 times ahead.
- However, the silica membranes were delivering a H₂/CO₂ selectivity of 770, which is 43 times higher than the graphene membranes.
- The point here is that overall the silica membrane and other similar membranes reported in the literature are ahead of the graphene membrane. Further, if the same silica membrane would deliver a H₂/CO₂ selectivity of 18 (instead of 770), then the H₂ permeance of the silica membrane would increase exponentially and not linearly.

10) Thanks and noted.

11) Thanks and noted.

12) Thanks and noted.

Finally, please note that this reviewer has never been part and never published papers with the Verweij's group.

Response to the reviewer's comments

Reviewer 3

The authors carried out a revision of their manuscript where many questions were adequately addressed. However, there are several points which still require clarification as set out below:

1) This reviewer is still concerned with the concept of “large area”. This concept is relative to graphene membranes only and it is not absolute to membrane areas conventionally reported for other graphene membranes, or to any other type of membranes. This concept may give the wrong impression that extremely low areas as in the case of this paper can be considered as large area for those working in membrane science and technology.

Author's response: We thank the reviewer for helping us in avoiding a confusion in the field. We used the term large-area because 1 mm² is 40000 fold higher than the 25 μm² area that has been the previous state-of-the-art, which we believe is a significant development in the field of single-layer graphene membrane.

Of course, we intend to develop graphene membranes that are m² in area. Keeping future developments in the single-layer graphene in mind, we have removed “large-area” from the title and the text. The title is now as following:

“Single-layer graphene membranes by crack-free transfer for gas mixture separation”

A section heading and a sentence in the discussion has been revised as follows:

Crack-free transfer of CVD graphene

“We developed a scalable NPC film-assisted transfer method to fabricate crack- and tear-free, millimeter- scale suspended single-layer CVD graphene films”

2) Thanks and the point made is agreed.

3) Thanks and the edits have addressed this question.

4) Thanks for adding more points to figure S6 (Robinson Plot). Please note that inorganic membranes are also extensively used for industrial applications. Further, the comparison to polymeric membranes is a weak argument in terms of separation, as polymeric membranes are the lower denominator (i.e. low selectivity). As graphene membranes are inorganic membranes, a solid comparison with inorganic membranes is warranted. Further, some of the results added in figure S6 are actually not the best results, and authors are still comparing their results against mid-level membrane performance and missed several important performance results. For instance, Verweij's group published a paper in Science (de Vos & Verweij, Science 1998, 5357, 1710-1711) reporting silica membranes with H₂ permeances 5 times higher than the membranes in this work, whilst delivering H₂/CH₄ selectivities of 770. There are many other excellent results and authors should improve this aspect of their paper.

Author's response: A comparison of separation performance from the intrinsic defects with only 0.025% porosity with the state-of-the-art membranes is not the key focus of this manuscript. The key developments in the manuscript are

- 1) Development of a novel transfer technique that for the first time enables crack-free transfer of relatively large area single-layer graphene.
- 2) Study of molecular transport from the intrinsic defects of graphene. Most importantly, we show that these intrinsic defects are hydrogen selective.
- 3) Finally, we developed an ozone-based functionalization and etching chemistry that can further tune the separation performance.

We agree that adding more data to the Robeson-type plot benefits the readership of Nature Communication. Accordingly, we have added the more comparison data including from the Verweij's group in Fig. S6. Although the performance of the reported graphene membranes falls short in

selectivity, future developments is expected to improve the performance (for example, by increasing the porosity from 0.025% to 2.5%, the H₂ permeance is expected to increase by 100 fold), reducing the actual cost per unit membrane area.

Fig. S6. A comparison of single-layer graphene films (all eight membranes) with a low-density of intrinsic defects (0.025%) with membranes in the literature in terms of the H₂/CH₄ separation performances. The red line is the polymer upper bound assuming 1 μm -thick skin layer¹¹, ZIF-8¹², silica¹³⁻¹⁹ and zeolite membranes (SSZ-13²⁰, AIPO-18²¹, DDR²², LTA²³, CHA²⁴).

5) Thanks for adding the explanation on the activated model. Could you please change the term translocation to diffusion? Translocation is generally used for protein transport through membranes (see Verner & Schatz, Science, 1988, 241, 1307-1313). Diffusion is the correct term used for the transport of gas through membranes. There is an immense amount of literature on gas diffusion for membranes, and using this correct term will be more consistent with the understanding of those readers and researchers working in membrane science and technology for gas separation. Further, the energy of activation is consistent with the term energy of diffusion of gas molecules.

Author's response: We have now replaced *translocation* by *diffusion*.

6) Thanks for citing the works from Barrer and Burggraaf. Please change “nanoporous graphene” to “microporous materials”. Please note that microporous materials are those with pore diameters below 2nm according to IUPAC, which is extensively used in studies about inorganic membranes complying with activated transport.

Author's response: For the sake of avoiding confusion, we have removed the term “nanoporous graphene” from the statement. The statement now reads as:

“To understand the transport behavior, the activation energy for gas diffusion across the nanopores was extracted from the temperature-dependent gas flux using an adsorbed phase transport model developed using the concepts of adsorption and diffusion³⁶⁻³⁹”.

7) Again, the paper quoted on experimental work of He adsorption on graphene was at extremely low temperature (liquid nitrogen temperatures). The point here is that if the authors carry out a He adsorption on graphite at the temperature range used in this work (25 to 200C), He adsorption is likely not to occur. In this case, the apparent energy of activation will be almost equal to the energy of diffusion of He through the membrane.

Author's response: In the previous round of review, we indicated that our own molecular simulation on adsorption of He yielded an adsorption energy close to 2 kJ/mole (molecular simulation method reported in the reference 37). 2 kJ/mole is a small adsorption energy, and as such the adsorption is poor. It is smaller than the thermal fluctuation, RT , which corresponds to 2.5 kJ/mole at 25 °C. It is also smaller than the error in fitting the activation energy, which is 3.2 kJ/mole. Moreover, the adsorption energy values between 0 and 2 kJ/mole do not change the conclusion from the observed molecular transport.

We have now revised the manuscript, using $E_{He,sur} = 0$ kJ/mol in the manuscript:

“Average E_{act} across eight membranes for He, H₂, CO₂, and CH₄ were 14.7 ± 3.2 , 20.2 ± 2.7 , 31.3 ± 2.8 , and 25.8 ± 4.8 kJ/mol, respectively.”

8) This is an interesting result which was not clear before, as gas selectivities were reported. High quality inorganic membranes complying with activated transport generally deliver a negative apparent activation energy for the transport of CO₂ (i.e. flux or permeance decreases with temperature). This is mainly due to the fact that the CO₂ heat of sorption is higher than the CO₂ energy of diffusion. In the case of this work, the opposite effect is reported as CO₂ has a positive apparent activation energy. Any comments please.

Author's response: The apparent activation energy is the sum of adsorption energy (negative term) and activation energy (positive term). The adsorption energy of CO₂ in graphene (17 kJ/mole) is smaller than that in zeolites and metal-organic frameworks (MOFs). For example, the isosteric heat of adsorption for CO₂ in Na-ZSM-5 has been reported to be 50 kJ/mole (*Langmuir* 1996, 12, 5896–5904).

9) This reviewer is still concerned with the concept of ultrahigh permeance. The permeances of the graphene membranes in this work are not high and the gas selectivities are low. This is not a criticism to the novelty of this work. However, it is a criticism to these statements which will not be accepted by those working in the membrane community. Let us compare the results in this paper with those published by Verweij's group in Science (de Vos & Verweij, Science 1998, 5357, 1710-1711). There are also a number of papers given similar performance as those published by Verweij's group. The arguments here are as follows:

- H₂ permeance were 5 times higher than this work and the silica membranes had a thickness of 30 nm.
- The permeance is inversely proportional to the thickness of the membrane. If the silica membrane could be produced with a thickness of 1 nm (like single graphene membranes), than silica membranes would have a permeance of 150 times higher than the graphene membranes in this work.
- This reviewer agrees that the porosity of the graphene membrane is low (0.025%). Silica membranes have porosities ranging from close to 0% all the way to 30%. Let us assume a high porosity value of 25% for the silica membranes. So the porosity of the graphene membranes is 1000 times lower.
- Then if we compare the permeances (150 times) over the porosity (1000 times), the graphene membrane will be a factor of 8 times ahead.

- However, the silica membranes were delivering a H₂/CO₂ selectivity of 770, which is 43 times higher than the graphene membranes.

- The point here is that overall the silica membrane and other similar membranes reported in the literature are ahead of the graphene membrane. Further, if the same silica membrane would deliver a H₂/CO₂ selectivity of 18 (instead of 770), then the H₂ permeance of the silica membrane would increase exponentially and not linearly.

Author's response: We thank the reviewer for explaining the difference in performance of the silica and graphene membranes. The term "ultrahigh permeance" was used to highlight the potential of graphene membrane as indicated by several simulations and not the reported membrane. However, to avoid the confusion, we have now revised the manuscript as follows:

"The single-layer nanoporous graphene film has the potential to be the ultimate membrane, yielding a high permeance and an attractive molecular selectivity."

10) Thanks and noted.

11) Thanks and noted.

12) Thanks and noted.

Reviewer's comments

Reviewer #3 (Remarks to the Author):

The authors addressed all my questions adequately. It is now recommended for the manuscript to be accepted for publication.